

# Mechanical/thermal sensitivity and superficial temperature in the stump of long-term tail-docked dairy cows

Renata J. Troncoso[1,*], Daniel E. Herzberg[2,*], Constanza S. Meneses[2,*], Heine Y. Müller[2,*], Marianne P. Werner[3,*] and Hedie Bustamante[1,*]

[1] Veterinary Clinical Sciences Department, Universidad Austral de Chile, Valdivia, Chile
[2] Graduate School, School of Veterinary Sciences, Universidad Austral de Chile, Valdivia, Chile
[3] Animal Science Department/School of Veterinary Sciences, Universidad Austral de Chile, Valdivia, Chile
[*] These authors contributed equally to this work.

Corresponding authors
Marianne P. Werner,
marianne.werner@uach.cl
Hedie Bustamante,
hbustamante@uach.cl

## ABSTRACT

**Background**. Tail docking of dairy cows is a painful procedure that affects animal welfare level. The aims of this study were first to evaluate the response to mechanical and thermal stimulation, and second to determine the superficial temperature of the stump of tail-docked dairy cows.

**Methods**. One hundred and sixty-four dairy cows were enrolled. From these, 133 cows were assigned to the tail-docked (TD) group and 31 cows were selected as control animals. The following sensory assessments to evaluate pain in tail-docked cows were performed. Sensitivity of the tail region in both groups of animals was evaluated using a portable algometer. Cold and heat sensitivity assessment was performed using a frozen pack (0 °C) and warm water (45 °C), respectively. Pinprick sensitivity was evaluated using a Wartenberg neurological pinwheel. Superficial temperature was evaluated using a thermographic camera. All sensory assessments and superficial temperature were evaluated in the ventral surface of the tail stump (TD) and tail (C).

**Results**. Pressure pain threshold was lower in TD cows (5.97 ± 0.19 kg) compared to control cows (11.75 ± 0.43 kg). Heat and cold sensitivity was higher in the TD cows compared to control cows with 29% and 23% of TD cows responding positively, respectively. Similarly, after pinprick sensitivity test was performed, 93% of TD cows elicited a positive response to stimulation. Tail-docked cows had lower superficial temperature (26.4 ± 0.27 °C) compared to control cows (29.9 ± 0.62 °C).

**Discussion**. Pressure pain threshold values in both groups of animals were higher than those previously reported for TD pigs, sows and cows. In contrast, pinprick stimulation evaluates the presence of punctate mechanical hyperalgesia/allodynia, usually related to traumatic nerve injury, and this association may reveal that it is possible that these animals developed a disorder associated to the development of a tail stump neuroma and concurrent neuropathic pain, previously reported in TD lambs, pigs and dogs. Thermal sensitivity showed that TD cows responded positively to heat and cold stimulation. These findings suggest that long-term TD cows could be suffering hyperalgesia/allodynia, which may be indicative of chronic pain. Lower superficial temperature in the stump may be associated to sympathetic fiber sprouting in the distal stump, which can lead to vasoconstriction and lower surface temperatures. Further studies are needed in order to confirm neuroma development and adrenergic sprouting.

## INTRODUCTION

Tail docking of dairy cows negatively impacts animal welfare (*Stull et al., 2002*). It comprises the removal of a part of the tail and is usually performed by applying a rubber or latex ring a few centimeters distal to the ventral aspect of the vulva (*Petrie et al., 1996*). Nowadays, tail docking is considered a controversial practice with few studies reporting improvements in udder and milk hygiene, cleaner cows by reducing the exposure to manure and mud and promoting personnel comfort during the milking process (*Schreiner & Ruegg, 2002a*; *Stull et al., 2002*; *Aubry, 2005*). In contrary, several reports have evaluated indirect measures of animal welfare and have found that docked cows have increased fly loads leading to alterations of eating patterns resulting in a decrease in milk production and increased fly avoidance behavior (*Phipps, Matthews & Verkerk, 1995*; *Eicher et al., 2001*) and restless behavior, including an increase in dorsal and lateral tail stump movements (*Eicher & Dailey, 2002*; *Tom et al., 2002*; *Eicher et al., 2006*). Similarly, other studies have not found differences in animal cleanliness, milk quality and somatic cell count between docked and non-docked animals (*Tucker, Fraser & Weary, 2001*; *Schreiner & Ruegg, 2002a*).

Tail docking is prohibited in countries like Denmark, Germany, Sweden, Scotland, England, Wales, and several states in the United States and Australia (*Hepple & Clark, 2011*; *AVA, 2012*; *AVMA, 2013*). Similarly, the American and Canadian Veterinary Medical Associations strongly oppose routine tail docking of cattle for management purposes (*CVMA, 2016*; *AVMA, 2013*). In Chile, although not currently forbidden, a marked decrease in its practice has been observed. Chilean legislation only indicates that painful procedures, such as tail docking should be performed in a manner that minimizes pain and suffering (*Chile, 2013*).

Several reports indicate that tail docking results in few behavioral and physiological signs of acute pain and distress in mature cows (*Petrie et al., 1996*; *Eicher et al., 2000*; *Tom et al., 2002*). Today, veterinarians and general public accept the notion that chronic pain is different from acute pain (*Reichling & Levine, 2009*); nonetheless, the uncertainty of whether acute pain can lead to the development of chronic pain still exists (*Voscopoulos & Lema, 2010*). According to *Flecknell (2008)*, the inconsistence of pain relief in cattle is the inadequate ability to assess pain. Chronic pain assessment has not been investigated thoroughly in cattle, but castration and tail docking may be associated to the development of chronic pain (*Molony & Kent, 1997*; *Eicher et al., 2006*). According to *Kroll et al. (2014)* there is an increased risk for potential chronic pain development at the amputation site, which has not been evaluated thoroughly in cows from commercial dairy farms. Quantitative sensory testing (QST) is usually performed in order to diagnose chronic pain conditions (*Cruz-Almeida & Fillingim, 2014*). It includes different methods that allow a characterization of somatosensory function or dysfunction. The most common methods include thermal (heat, cold) and mechanical (tactile, pressure) stimulation in order to elicit a painful or nonpainful response (*Fillingim et al., 2016*). In addition, skin

temperature evaluation can help determine tissue metabolism and blood circulation; therefore, changes could reflect circulatory or inflammatory conditions associated to chronic pain (*Sathiyabarathi et al., 2016*).

The objectives of this study were first to evaluate the response to mechanical and thermal stimulation, and second to determine the superficial temperature of the stump of tail-docked (TD) dairy cows.

## MATERIALS & METHODS

### Animals and housing

This study was conducted between November and December 2015 on a commercial farm located in Los Rios Region, southern Chile. The study was approved by the Ethics and Bioethics Committee of Animal Research of the Universidad Austral de Chile (MV.21.2015). A total of 164 Holstein dairy cows with a mean age of 6.2 $\pm$ 1.9 years (parity range: 3–4), mean body weight of 423 $\pm$ 26 kg, mean milk yield of 27.3 $\pm$ 5.4 L day$^{-1}$ were enrolled. Only cows without clinical signs of systemic disease, mastitis or lameness during the last 15 days were selected. All evaluated cows were housed individually in a tie-stall, fed a total mixed ration (TMR) and milked three times a day during the entire period of study. From these, 133 cows were assigned to the TD group and 31 cows were selected as control (C) animals and identified using the ear tag farm number. Individual register showed that cows in the TD group were tail-docked at a mean age of 11.9 month (range = 11.7–12.4) using a rubber band at a distance of approximately 10 cm below the vulva by the farm veterinarian.

### Study design

A clinical quantitative sensory assessment protocol was developed in order to evaluate the presence of pain in TD cows. Prior to sensory testing, cows were habituated to the presence of the evaluator and experimental testing was performed during three consecutive days. After the morning milking, cows were allowed to return to their tie-stall individual cubicles and were restrained using a headlock self-locking system for sensory assessment. The same evaluator (RT) performed all the sensory assessments with the assistance of another researcher in charge of identifying and recording positive reactions to the sensory stimuli (*Eicher et al., 2006*). In order to avoid stress in the animals, both researchers approached the animals in a calm and quiet manner.

### Sensory assessments

None of the animals received analgesic treatment prior to the sensory evaluation. The following tests were performed.

Pressure pain sensitivity: Sensitivity of the tail region in both groups of animals was evaluated using a portable algometer (Wagner FDX 25 Compact Digital Force Gauge, Wagner Instruments, Riverside, CT, USA) with a 1 cm$^2$ rubber probe. For each evaluation, the probe was constantly applied in the same topographical location and placed perpendicular to the skin. The amount of pressure applied during each evaluation was constantly increased at 500 g of force per second in the ventral surface of the tail stump (TD), and ventral surface of the tail (C), respectively, until a positive response was

 

obtained. Each area was assessed five times at 60-s intervals. Lateral and ventral movement and/or withdrawal of the tail were considered positive responses, in which the pressure elicited by the algometer was immediately discontinued and pressure registered. The mean of five measurements per site was considered as a single value per tested cow.

Thermal sensitivity: Cold and heat sensitivity assessment was performed using a frozen pack (0 °C) and warm water (45 °C), respectively. Both stimuli were applied for 5 s in the ventral surface of the tail stump (TD) and ventral surface of the tail (C), respectively, or until a positive response was obtained. Lateral and ventral movement and/or withdrawal of the tail were considered positive responses.

Pinprick sensitivity: The pinprick sensitivity was evaluated using a Wartenberg neurological pinwheel applied in the ventral surface of the tail stump (TD) and ventral surface of the tail (C), respectively. Lateral and ventral movement and/or withdrawal of the tail were considered positive responses.

## Superficial temperature

Superficial temperature was evaluated using a thermographic camera (FLIR® i5, Wilsonville, OR, USA) calibrated with an emissivity ($\varepsilon$) of 0.95 according to the manufacturer. Images from the ventral surface of the tail stump (TD) and ventral surface of the tail (C) were obtained at a distance of 10 cm. All images were obtained before sensory stimuli were applied. Thermogram analysis was performed using the FLIR® Tools 5.4 software (FLIR Systems Inc., Wilsonville, OR, USA), and atmospheric temperature and relative humidity were included in the analysis. To come to a single representative value, the mean temperature obtained from five longitudinal lines along the ventral surface of the tail, was considered.

## Statistical analysis

For each continuous variable, probability plots were generated to verify that data followed a normal distribution. Pressure pain threshold and superficial temperature were analyzed using analysis of covariance. The adjusted linear model included condition (TD versus C) as fixed effect and age as covariate. Body weight, parity and milk yield were not associated with pressure pain threshold and superficial temperature, and thus were removed from the model using a backward selection process. In order to analyze for a possible association between condition and sensitive stimulation, Pearson's Chi-square were conducted for pinprick and heat sensitivity and Fisher's Exact test was conducted for cold sensitivity. For all statistical procedures, the overall alpha was set to 0.05. The statistical analysis was performed using R Statistical Software (*R Development Core Team, 2018*).

## RESULTS

Lower pressure pain threshold values were necessary to obtain a withdrawal response in TD cows (5.97 $\pm$ 0.19 kg) compared to C cows (11.75 $\pm$ 0.43 kg) ($P < 0.001$, $\eta_p2 = 0.46$) (Fig. 1). Condition (TD versus C) was associated with heat sensitivity ($\chi^2 = 10.36$, $df = 1$, $P = 0.001$) with 29% of TD cows responding positively (Table 1). Also, condition (TD versus C) was associated with cold sensitivity ($P = 0.04$) with 23% of TD cows responding

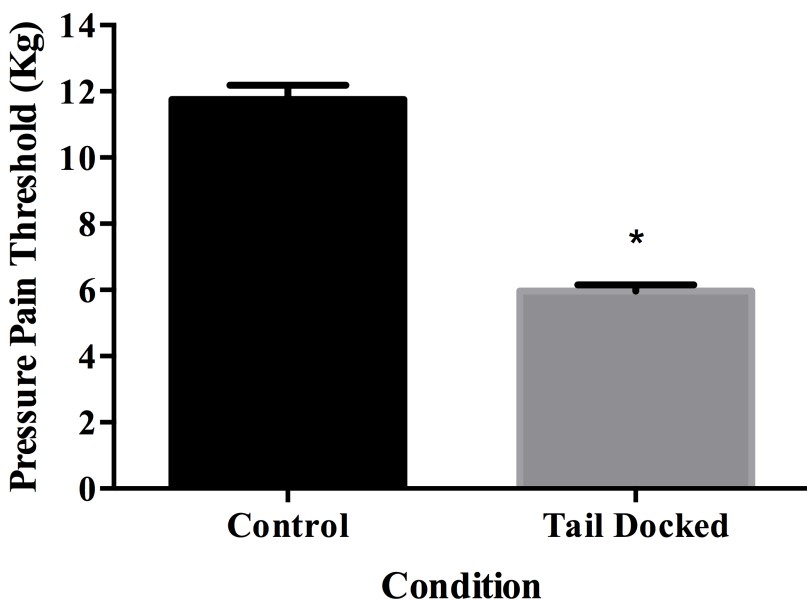

**Figure 1** Least square means and standard error for pressure pain threshold in tail docked ($n = 133$) and control cows ($n = 31$). *Statistically significant differences between groups ($P < 0.001$).

**Table 1** Frequencies and percentages of sensory assessment in tail-docked ($n = 133$) and control cows ($n = 31$).

|  | Tail-docked | | Control | | P-value |
|---|---|---|---|---|---|
|  | Positive N (%) | Negative N (%) | Positive N (%) | Negative N (%) |  |
| Heat sensitivity | 39 (29) | 94 (71) | 0 (0) | 31 (100) | 0.001[*] |
| Cold sensitivity | 31 (23) | 102 (77) | 2 (7) | 29 (94) | 0.04[**] |
| Pinprick stimulus | 124 (93) | 9 (7) | 23 (74) | 8 (27) | 0.005[*] |

**Notes.**
[*]P-values for Chi square test.
[**]P-value for Fisher's exact test.

positively (Table 1). Similarly, after pinprick sensitivity test was performed, 93% of TD cows elicited a positive response to stimulation. This sensory testing was associated with the condition ($\chi^2 = 7.87$, $df = 1$, $P = 0.005$). TD cows had lower superficial temperature ($26.4 \pm 0.27\,°C$) compared to C cows ($29.9 \pm 0.62\,°C$) ($P < 0.001$, $\eta_p2 = 0.13$) (Fig. 2).

# DISCUSSION

Painful procedures are performed in the dairy industry and they are often associated with the development of fear, distress and chronic pain of animals (*Grandin, 2015*). Tail docking is a painful procedure that induces both acute and chronic pain, and leads to behavioral modifications and discomfort (*Tucker, Fraser & Weary, 2001*). Different studies have confirmed the presence of acute pain and augmented animal activity, characterized by a marked increase in foot stomp behavior following tail docking (*Eicher & Dailey, 2002*; *Schreiner & Ruegg, 2002b*; *Tom et al., 2002*).

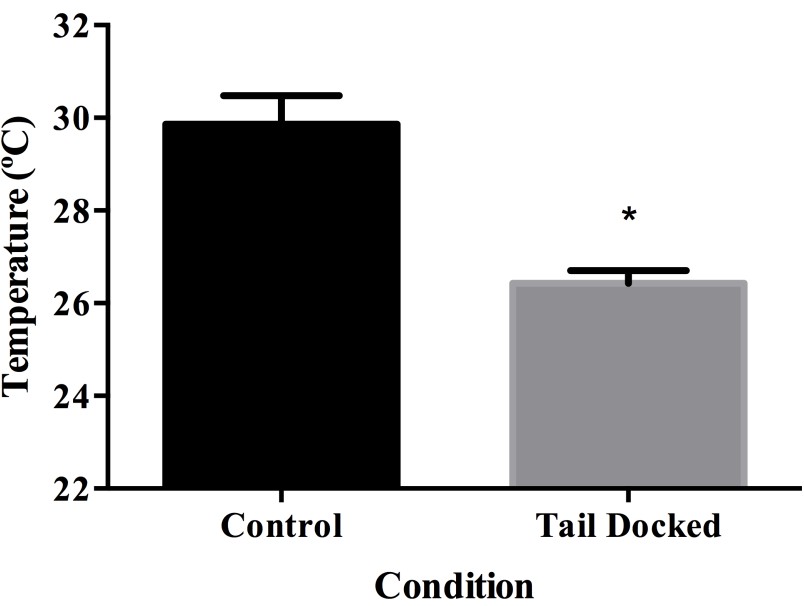

**Figure 2** **Least square means and standard error for superficial temperature in tail docked ($n = 133$)** **and control cows ($n = 31$).** *Statistically significant differences between groups ($P < 0.001$).

Tail-docked cows had lower pressure pain threshold compared to controls. These results are similar to those reported in pigs, in which mechanical sensitization of the tail stump lasted for up to 16 weeks (*Di Giminiani et al., 2017*). Pressure pain threshold values in both groups of animals were higher than those previously reported for TD pigs (*Di Giminiani et al., 2016*), sows (*Nalon et al., 2016*) and cows stimulated using an algometer with a metal probe in the third metatarsal bone (*Raundal et al., 2014*). The higher overall pressure values described in this study could be related to the use of a rubber probe. According to *Di Giminiani et al. (2016)*, the use of different probes could be associated to an increased degree of response variability. Similarly, *Taylor & Dixon (2012)* mention that an increase in probe diameter results in higher variability. Other factors that may influence the higher values of pressure threshold presented here may include skin thickness (*Di Giminiani et al., 2016*), individual variation (*Nalon et al., 2016*), and stress-induced hypoalgesia (*Herskin, Munksgaard & Ladewig, 2004*).

The association between condition and heat sensitivity is similar to that reported by *Eicher et al. (2006)*, in which TD cows manifested less foot stomps, foot shifts and tail swings. A positive response to cold stimulation after tail docking was previously reported by *Eicher et al. (2006)*, with TD cows showing an increased number of foot stomps after $-9\ ^{\circ}\text{C}$ cold stimulation. Similarly, an increase response to cold stimulation at $0\ ^{\circ}\text{C}$ has been described in amputated human patients diagnosed with phantom limb pain (*Li, Melton & Li, 2015*). Moreover, here we report a significant association between condition and pinprick sensitivity. Impaired sensitivity to pinprick has been previously reported in amputated human patients (*Kosasih & Silver-Thorn, 1998*). Pinprick stimulation evaluates the presence of punctate mechanical hyperalgesia/allodynia, usually related

to traumatic nerve injury (*Jensen & Finnerup, 2014*). This association suggests that animals may have developed tail stump neuroma as reported previously in other species. *Petrie et al. (1996)* indicate that tail docking would induce tissue damage that leads to neuromata development and concurrent neuropathic pain. Moreover, neuroma development has been previously reported in tail-docked lambs (*French & Morgan, 1992*; *Fischer & Gregory, 2007*), pigs (*Herskin, Thodberg & Jensen, 2015*; *Kells et al., 2017*) and dogs (*Gross & Carr, 1990*). Peripheral neuromas occur in 10–25% of human amputees, and are generally formed after injury or surgical procedures, resulting in neuropathic pain, residual limb pain, functional impairment and psychological distress (*Rajput, Reddy & Shankar, 2012*), increasing sensitivity to mechanical and thermal stimulation (*Toia et al., 2015*; *O'Reilly et al., 2016*; *Yao et al., 2017*). Histopathological analysis confirmed the presence of neuroma in the tail stump of docked pigs one month after tail docking, characterized by marked nerve sheath and axonal proliferation (*Sandercock et al., 2016*). Moreover, another study in pigs identified age at time of the procedure as a factor that may influence the development of neuropathic pain (*Di Giminiani et al., 2017*). Nonetheless, cows in the present study were, on average tail docked 48 months before sensory evaluation. According to this, we believe that pain experienced by docked cows is similar to human phantom limb pain (*Nikolajsen, 2012*). In this condition, the amputation and trauma that nerves and surrounding tissue suffer, disrupts normal afferent and efferent signals, leads to neuroma development and neurons become hyper-excitable (*Hanyu-Deutmeyer & Dulebohn, 2018*). Phantom limb pain has been vastly studied in humans (*Schley et al., 2008*; *Andoh et al., 2017*; *Yin et al., 2017*). In cases of phantom limb pain, characteristic chronic neuropathic pain occurs in the amputation stump; and although this pain may decrease or eventually disappear over time, if it continues for more than 6 months, the prognosis for pain decrease is poor (*Kuffler, 2017*).

Surface temperature was significantly lower in the TD group compared to controls. Similar results were reported by *Eicher et al. (2006)*, where the stump of docked heifers was approximately 2 °C colder than the underside of the tails of intact heifers. Similar results have been described in amputated humans, in which the stump of amputated limbs had lower superficial temperatures than the contralateral side using a temperature probe (*Hunter, Katz & Davis, 2005*) and thermographic analysis (*Harden et al., 2008*). This decrease in temperature may be associated with sympathetic fiber sprouting in the distal stump, which can lead to vasoconstriction and lower surface temperatures (*Harden et al., 2004*). Similarly, *Nascimento et al. (2015)*, after traumatic nerve injury confirms the presence of sympathetic sprouting in the skin that contributes to pain.

In this study, we showed evidence that may confirm the development of chronic pain states (hyperalgesia/allodynia) in long term TD dairy cows. The principal limitation of this study is the unequal number of animals in the two experimental groups, in which the lower number of control cows compared to the TD group could have influenced the results. Future research in TD cows must include the evaluation of other indicators of welfare such as behavior, motor activity and plasma biomarkers of pain and stress. Moreover, in order to confirm neuropathic pain, neuroma formation in cows must be demonstrated using a thorough histopathological examination. Nonetheless, our results confirm that tail docking of dairy cows is a practice that affects animal welfare (*Stull et al., 2002*).

## CONCLUSIONS

Tail-docked cows had an increased response to mechanical stimulation characterized by lower pain pressure thresholds and a positive association to pinprick sensitivity. Thermal sensitivity showed that TD cows responded positively to heat and cold stimulation. These findings suggest that in the long-term TD cows could suffer from hyperalgesia/allodynia, which may be indicative of chronic pain. Lower superficial temperature in the stump could be associated with adrenergic tissue sprouting inducing peripheral vasoconstriction. Further studies are needed in order to confirm neuroma development and adrenergic sprouting.

### Funding

The authors received no funding for this work.

### Competing Interests

The authors declare there are no competing interests.

### Author Contributions

- Renata J. Troncoso conceived and designed the experiments, performed the experiments, analyzed the data, authored or reviewed drafts of the paper, approved the final draft.
- Daniel E. Herzberg and Constanza S. Meneses conceived and designed the experiments, analyzed the data, authored or reviewed drafts of the paper, approved the final draft.
- Heine Y. Müller conceived and designed the experiments, analyzed the data, contributed reagents/materials/analysis tools, prepared figures and/or tables, authored or reviewed drafts of the paper, approved the final draft.
- Marianne P. Werner and Hedie Bustamante conceived and designed the experiments, performed the experiments, contributed reagents/materials/analysis tools, prepared figures and/or tables, authored or reviewed drafts of the paper, approved the final draft.

### Animal Ethics

The following information was supplied relating to ethical approvals (i.e., approving body and any reference numbers):

The study was approved by the Ethics and Bioethics Committee of Animal Research of the Universidad Austral de Chile (MV.21.2015).

### Data Availability

The raw data used to generate the R data frame are provided in Supplemental File.

### Supplemental Information

Supplemental information for this article can be found online at http://dx.doi.org/10.7717/peerj.5213#supplemental-information.

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
