# Peer review of "Mechanical/thermal sensitivity and superficial temperature in the stump of long-term tail-docked dairy cows"

_PeerJ, doi:10.7717/peerj.5213_

## Round 0.1 · original submission · Major Revisions

All three reviewers (who all did a thorough review) saw merit in your study and manuscript. However, they all have concerns. I agree with their comments so please respond to them all in detail.

Reviewer 1 ·

Basic reporting

This is a very well written and important study, as tail-docking of dairy cattle is an important animal welfare problem in the dairy industry across the globe. This is a study that can be useful to the field if properly discussed, especially its limitations. Please re check the order of the values presented and please include a discussion about the meaning of the results found and applicability of the results to the care of dairy cattle..

Experimental design

1- M&M: Missing important details to the full comprehension of the study.
This study seems to be performed appropriately, but the materials and methods are lacking information.

Validity of the findings

Interesting discussion about pain and sensitivity around amputated limbs and tails in other animals. However, the introduction is not connected to the introduction. Also, it is missing a discussion on a major part of the meaning of the results found and applicability of the results in the care of dairy cattle.

Additional comments

Title : Please include an space between Mechanical and thermal.
Line 13: please modify for “ Animal Welfare level”.
Line 14: Incomplete sentence
Line 55: This is an old citation. Many other countries have banned this practice since 2011, such as the USA.
Line 68: Same comment
Line 80: Please include more details about the housing, and the tail-docking procedure. Details such as, who did it? Location on the tail where the docking was performed? And etc.
Methodology: These are common tests that have been performed before, please include citations for the methods used and explain the adaptation made.
Line 124: There are many other co-variables that could influence this model, such as place of docking, and cow body weight. Why just age was included in the model? Also, which Chi-square test was performed?
Line 161: New paragraph.

·

Basic reporting

The authors performed a well designed study with clear and significant results. Although there was some individual variation in the reaction of the cows, the conclusions are that tail docking has consequenses for the welfare of the cows involved.

The presentation of the findings are clear and the available literature is included in the discussion of the results.

Just a few minor (spelling) comments:

150 Tail-docked cows showed significantly less pressure pain threshold compared to controls
Tail-docked cows showed significantly LOWER pressure pain threshold compared to controls

157 associated to an increase degree of response variability
associated to an increaseD degree of response variability

159 Another factors that may influence the higher values
Other factors that may influence the higher values

161 Start new paragraph

188 over time, if continues for more than 6 month,
over time, if IT continues for more than 6 monthS,

200 tail-docked cows had an increase response
tail-docked cows had an increaseD response

Experimental design

The authors performed a well designed study with clear and significant results.

Validity of the findings

The authors performed a well designed study with clear and significant results.

Additional comments

The authors performed a well designed study with clear and significant results. Although there was some individual variation in the reaction of the cows, the conclusions are that tail docking has consequenses for the welfare of the cows involved.

The presentation of the findings are clear and the available literature is included in the discussion of the results.

Just a few minor (spelling) comments:

150 Tail-docked cows showed significantly less pressure pain threshold compared to controls
Tail-docked cows showed significantly LOWER pressure pain threshold compared to controls

157 associated to an increase degree of response variability
associated to an increaseD degree of response variability

159 Another factors that may influence the higher values
Other factors that may influence the higher values

161 Start new paragraph

188 over time, if continues for more than 6 month,
over time, if IT continues for more than 6 monthS,

200 tail-docked cows had an increase response
tail-docked cows had an increaseD response

Reviewer 3 ·

Basic reporting

Introduction and Background.
This section could use some additional work. The quality of the writing in this section contrasts significantly with the quality of the writing in the discussion. The justification for the study is poorly developed and in terms of the objective of measuring superficial temperature is not developed at all.
There appears to be some confusion in the first paragraph around the benefits vs the welfare issues around docking. Perhaps the intro could be restructured to identify tail docking as a controversial practice with few studies supporting claims of improved cleanliness and disease to the point that many countries and organizations such as the AVMa coming out against the practice. The tone of the current introduction is quite the opposite
The novely of the study is limited to the location. Such a study may not have been done on a commercial farm in Chili previously but similar work has been done in research facilities in the US. (Eicher and Chang’s work)
The superficial temperature description in the materials and methods section could be move to the beginning of this section as it is done before the other tests.
Line 48 – “removal of an important part of the tail” is not very helpful in describing what is actually done. Perhaps the authors could describe how much of the lower part of the tail is usually removed.
Line 51 change different studies to other studies
Line 57problem with the sentence here. “ and disbudding should be performed minimizing” …could be changed to “ should be performed in a manner that minimizes…” but the Chile reference should be checked so the sentence is accurate.
Line 60-61 delete “ with the consequently” and replace with “ resulting in a “
Line 62 “ Contradictory evidence…” makes no sense here. I am not sure what the authors are trying to say. Perhaps expand the thought.
Line 68-69 the aim of determining the superficial temperature of the stump Is not justified in the intro
Line 72 “ change in a commercial farm to on a commercial farm
Line 73: delete “ with confined system”
Line 88 change to “ I n charge of identifying and recording positive reactions…”
Line 92 delete “previous” and replace with “prior to” or “before”
Line 116 change stimulus to stimuliLine 147-148 already stated in introduction delete
Line 169-170 Change “ this association may reveal that it is possible that these animals developed a disorder associated to the development of a tail stump neuroma” to “this association suggest that animals may have developed tail stump neuromas as reported previousl and in other species (refs)
I believe work by Cheng and Eicher has already demonstrated this finding.
Line 188 missing word. Should read “ if it continues” not “if continues”
Line 191 change “ dock heifers had” to dock heifers were

Experimental design

Not clear in materials and methods how long after tail docking the study was performed. Mean and range should be reported.
The document reads as if the researchers were able to assign animals to different treatments it is unclear why a 133 to 31 distribution was selected.
The only criteria mentioned for animal selection is the lack of disease. Were the two groups blocked for other characteristics, parity, age, size, breed etc.
Not clear in materials and methods about how frequently, which days, after docking were the measurements taken.
Numerous places in the materials and methods section the authors clearly identify where measures are taken for the TD animals but provide no information about where on the tail measures are taken for the control group. For example line 99 . Similarly, for temperature not clear if temperatures were taken in similar locations for both groups.

Validity of the findings

Very limited – one paragraph with minimal novelty
Very limited analysis in table 1
Data in figures is well presented and analyzed

---

## Round 0.2 · Minor Revisions

Please address the minor comments of Reviewer 1.

Reviewer 1 ·

Basic reporting

This is a very well written and important study, as tail-docking of dairy cattle is an important animal welfare problem in the dairy industry across the globe. In my opinion, the review form of this paper addresses most of the reviewers concerns. I believe the authors did a great job in addressing the comments from the three reviewers. The limitations of the study are now presented but it requires a more detailed discussion of the factors that could have been improved in this study. I have just a couple of comments to add.

Experimental design

This is a well-performed experiment, the revised version of the manuscript elucidated most of my questions about the methods used.

Validity of the findings

Tail-docking of dairy cattle is an important animal welfare problem in the dairy industry across the globe. This research is an important contribution to the long-term effects of the tail amputation.

Additional comments

This is a very well written and important study, as tail-docking of dairy cattle is an important animal welfare problem in the dairy industry across the globe. In my opinion, the review form of this paper addresses most of the reviewers concerns. I believe the authors did a great job in addressing the comments from the three reviewers. The limitations of the study are now presented but it requires a more detailed discussion of the factors that could have been improved in this study. I have just a couple of comments to add.

Minor comments:


Title : The modifications to the title did not make sense. I suggest: Long-term mechanical/thermal sensitivity and superficial temperature in the stump of tail-docked dairy cows.
Line 95: Please, change “modalities” to “methods”.
Line 126: Is there a reference for this methods or at least some parts of it.
Line 166: include the criteria for backwards elimination.
Line 169: If not the standard package, please include the name and reference of the package used.
Line 176: P values should be reported at maximum of 3 decimal places
Line 231: Please explain the mechanism associated with how this relate to the results found, also include the future studies that could be done in this topic.
Line 244: Which other factors? Also, please expand how the animals and the methods could be improved.
Please include a future research paragraph or throughout the discussion. This is not the last study that should be done in this topic.

·

Basic reporting

All the points raised in my first review are addressed.

Experimental design

All the points raised in my first review are addressed.

Validity of the findings

All the points raised in my first review are addressed.

Additional comments

All the points raised in my first review are addressed.

---

## Round 0.3 · Minor Revisions

When reading the last version of the manuscript, I detected too many typos and sections that need to be edited. My edits are of course suggestions. Before submitting the (hopefully) final version, please send it to my personal e-mail (barkema@ucalgary.ca) for me to advise you. Also, feel free to inquire why I made a suggestion to change or edit.

The system doesn't allow sending Word files. Staff will therefore send the Word document including Track Changes to the corresponding author.

---

## Round 0.4 · accepted · Accept

Thank you for making the changes in the manuscript. It is now ready to be published!

#